# Modulation of Muscarinic Signalling in the Central Nervous System by Steroid Hormones and Neurosteroids

**DOI:** 10.3390/ijms24010507

**Published:** 2022-12-28

**Authors:** Ewa Szczurowska, Eszter Szánti-Pintér, Nikolai Chetverikov, Alena Randáková, Eva Kudová, Jan Jakubík

**Affiliations:** 1Institute of Organic Chemistry and Biochemistry, Czech Academy of Sciences, Flemingovo Náměstí 2, Prague 6, 166 10 Prague, Czech Republic; 2Institute of Physiology, Czech Academy of Sciences, Vídeňská 1083, 142 20 Prague, Czech Republic

**Keywords:** neurosteroids, neuroactive steroids, cholesterol, muscarinic receptors, Alzheimer’s disease, Parkinson’s disease, schizophrenia, substance abuse, depression

## Abstract

Muscarinic acetylcholine receptors expressed in the central nervous system mediate various functions, including cognition, memory, or reward. Therefore, muscarinic receptors represent potential pharmacological targets for various diseases and conditions, such as Alzheimer’s disease, schizophrenia, addiction, epilepsy, or depression. Muscarinic receptors are allosterically modulated by neurosteroids and steroid hormones at physiologically relevant concentrations. In this review, we focus on the modulation of muscarinic receptors by neurosteroids and steroid hormones in the context of diseases and disorders of the central nervous system. Further, we propose the potential use of neuroactive steroids in the development of pharmacotherapeutics for these diseases and conditions.

## 1. Introduction

Besides their well-known genomic effects, neurosteroids (NSs) produced by neurons and glia modulate neuronal activity by binding to various membrane proteins. Their neuromodulatory effects vary depending on location and time. Among well-studied receptors modulated by neurosteroids are, for example, γ-aminobutyric acid (GABA_A_) [1] and the glutamate N-methyl-D-aspartate (NMDA) receptors [2]. In this review, we focus on less-studied modulation of muscarinic acetylcholine receptors by NSs and steroid hormones (SHs) in the central nervous system (CNS) and its implications in CNS-related disorders and diseases.

Muscarinic acetylcholine receptors (mAChRs) are G-protein coupled receptors (GPCRs) abundant throughout the whole body. All five subtypes of the muscarinic receptor are expressed in the CNS, where depending on their location and subtype mediate various functions, including cognition, memory, or reward. Therefore, mAChRs may serve as targets for various diseases and conditions such as Alzheimer’s disease, schizophrenia, addiction, epilepsy, or depression [3]. The orthosteric binding site of all five subtypes is identical [4,5,6,7], making it an unsuitable pharmacological target. In contrast, allosteric binding sites are not under the evolutionary pressure to accommodate natural neurotransmitters and thus vary in their structure, being a more suitable pharmacological target than the orthosteric site. Indeed, achievable selectivity, the advantage of allosteric modulators is the preservation of spatial-temporal pattern of the modulated signalling as positive allosteric modulators (PAMs) increase whereas negative allosteric modulators (NAMs) diminish or block responses to the neurotransmitter upon its release. Allosteric modulators may change both affinity and efficacy of the neurotransmitter as well as alter coupling to the downstream signalling pathways [8,9].

Thanks to strong interest and intensive research, the number of allosteric modulators of mAChRs is growing. Allosteric modulators of mAChRs encompass a broad spectrum of chemotypes [10]. Moreover, similar to many other GPCRs, mAChRs are allosterically modulated by membrane cholesterol (CRL) [11,12]. Some SH and NS, such as progesterone (PROG) or corticosterone, at physiologically relevant concentrations allosterically modulate the binding and functional response of mAChRs from the CRL-binding site [13,14]. These findings make neuroactive steroids interesting candidates for allosteric modulators of a novel scaffold. Here reviewed modulation of mAChRs by SHs and NSs under physiological and pathological conditions gives hints on desired properties of steroid-based allosteric modulators with pharmacological potential.

## 2. Neurosteroids and Neuroactive Steroids in CNS

NSs represent a class of endogenous compounds synthesized de novo in the CNS from CRL or steroidal precursors imported from peripheral endocrine glands. NSs are known to modulate neuronal signalling [12]. NSs and SHs exert multiple functions via two mechanisms. By the activation of nuclear receptors specific for a given steroid hormone they execute their genomic effects. Alternatively, by binding to the membrane proteins and modulating the activity of these proteins they execute their non-genomic effects. In this review, we will mainly focus on NSs and SHs that have been extensively studied over the last two decades—progesterone (PROG), pregnenolone (PREG), pregnenolone sulphate (PREGS), dehydroepiandrosterone sulphate (DHEAS), 3α-hydroxy-5α-pregnan-20-one (allopregnanolone, ALLO), 3α-hydroxy-5β-pregnan-20-one (pregnanolone), corticosterone, and 17β-estradiol.

The plasma level of circulating steroids is in the low nanomolar range [15]. Circulating steroids serve as precursors for the synthesis of neurosteroids [16]. However, it should be noted that effective concentrations of locally synthesised neurosteroids modulating membrane receptors including mAChRs may exceed the concentrations of circulation steroids [17].

## 3. Muscarinic Receptors in CNS

Muscarinic receptors are expressed throughout the entire CNS. Based on their subtype and location, they mediate a wide array of physiological processes and thus become pharmacological targets in various pathologies related to altered muscarinic signalling (Table 1). The M_1_ receptors are predominantly expressed in the neocortex, hippocampus, and striatum [18,19]. Stimulation of M_1_ signalling facilitates learning and memory [20]. Impaired M_1_ signalling constitutes the basis for cognitive decline and dysfunction in various forms of dementia including AD and cognitive symptoms of schizophrenia [21,22,23,24,25]. Overstimulation of M_1_ receptors may lead to seizures and epilepsy [26,27,28]. The M_2_ receptors are primarily expressed in the hippocampus, cortex, basal forebrain, and thalamus [19]. Their activation leads to neural inhibition and analgesia and also to tremors and hypothermia [26]. Inhibition of M_2_ receptors, for example, leads to increased locomotion and decreased food intake. The M_2_ receptors are implied in the management of pain [29] and depression [30]. The M_3_ receptors are mainly expressed in the peripheral nervous system. In the CNS, their expression is limited to the cortex and basal ganglia [19]. Similarly to M_1_ receptors, they improve learning and memory [31]. The M_4_ receptors are primarily expressed in the striatum, neocortex, hippocampus, and basal ganglia [19,24,32]. Similarly to M_2_ receptors, their activation leads to auto-inhibition of acetylcholine (ACh) release and analgesia [33]. Additionally, it decreases the release of dopamine in the striatum [34] controlling rhythmicity of locomotor activity [35] and having anti-psychotic effects [36]. The M_4_ receptors are proposed pharmacological targets in schizophrenia [21,24,25,37,38], Parkinson’s disease (PD) [34,39], and pain management [40]. In comparison to other mAChRs, the expression of M_5_ receptors is scarce and is limited mainly to the ventral tegmental area (VTA), substantia nigra (SN), and brain microvasculature [19,24,31] and to a lesser extent to cerebellum [41]. Activation of M_5_ receptors in microvasculature results in cerebral vasodilation. The VTA is a part of the mesocorticolimbic circuit that works as a reward system. Activation of M_5_ receptors in the VTA induces dopamine release [26,27], promoting drug-seeking behaviour and reward [25]. The M_5_ receptors are proposed pharmacological targets in AD [42,43], schizophrenia [25], and substance abuse [43,44]. For further details see Figure 1 and Table 1.

## 4. Genomic Effects of Steroids on mAChRs in CNS

Effects exerted by steroids that take place over a longer timescale include genomic action [48]. This accounts for 17β-estradiol and PROG as it represents a fundamental effector in the regulation of the menstrual cycle and cognition in females. Age-related decreases in the 17β-estradiol as well as PROG, 5α-Dihydroprogesterone (5α-DHP), and ALLO levels within CNS in older age [49] are thought to be implicated in cognitive decline and higher risk of Alzheimer’s disease (AD) [50,51,52,53]. These steroidal changes in men are associated with alterations in mAChRs expression levels within the brain. 

There are numerous studies on mAChRs expression changes in the human brain, however, the results are often unmatched. Some of the analyses conducted on post-mortem brain tissue described marked decline [54], while other studies show only minor changes in the overall mAChRs population detected in ageing individuals [55]. The results of the more detailed in vivo positron emission tomography (PET) studies in age-related muscarinic decline brought even more contradictions. Both reductions [56,57,58,59] and increases [60] in the densities of mAChRs were reported. These inconsistencies are mainly due to variability in the populations (males and females), selectivity of used ligands, and differences in quantification methods. To avoid such discrepancies [61], the age-related differences between M_1_ and M_4_ receptors in the population of healthy women have been examined, employing the single photon emission tomography (SPET) of the M_1_/M_4_ selective probe (R, R)[^123^I]-I-QNB. This study revealed an age-related reduction in mAChRs densities.

To date, many animal and human studies revealed that Estrogen Therapy (ET), if initiated within a specific time window, can bring beneficial effects on cognition and reduce the risk of dementia by enhancing the cholinergic system via increasing choline acetyltransferase (ChAT) activity as well as high-affinity choline uptake [51,62,63]. In animals, mAChRs densities fluctuate during oestrous cycles and it is positively correlated with the levels of estrogens. During the proestrus characterized by the highest levels of estrogen, the mAChRs densities increase, while during diestrous, the mAChRs densities decline together with estrogens levels [64]. It suggests that estrogens regulate mAChRs expression levels by their genomic action.

Ovariectomy is often used in laboratory animals as a model of menopause. It was shown that the removal of ovaries is associated with a decrease in ChAT mRNA and changes in the expression of mAChRs and a decrease in cholinergic fibre density [65,66]. Experimental studies in ovariectomized rats suggested that estrogen level affects mAChRs expression in the cerebral cortex [64], medial basal hypothalamus [67,68], and hippocampus [69,70,71]. Interestingly, a recent study did not find any changes in the density of mAChRs in the brain of ovariectomized rats [72]. 

Changes in mAChRs expression levels reported across mentioned studies varied, depending on the time post-ovariectomy, the time when the 17β-estradiol replacement was initiated, the 17β-estradiol dose as well as on the time point at which expression was measured. In general, the removal of ovaries resulted in an initial, significant increase in mAChRs expression levels throughout the brain which might be contracted by 17β-estradiol replacement (Table 2). On the contrary, some researchers reported a marked decrease in mAChRs expression levels in mice brains following ovariectomy [65]. In this study, the authors showed that 17β-estradiol-induced improvement in cognitive performance was not correlated with ovariectomy itself nor with the levels of mAChRs, which remained unchanged after 17β-estradiol treatment. Hormonal replacement therapy often comprises a combination of 17β-estradiol and progesterone. In the reported animal studies, 17β-estradiol and progesterone [64] or progesterone alone [63] did not influence mAChRs expression levels. On the contrary, some studies suggest that progesterone may counteract the effects of estrogens [73].

As for human studies, there are very few publications revising estrogen-induced changes in mAChRs expression levels. An example might be the single-photon emission tomography (SPET) study reporting higher mAChRs brain densities in postmenopausal women receiving estrogen therapy [74].

The exact mechanisms and relationship between the changes in the densities of mAChRs and estrogens level are still not fully understood, however, some concepts may partially explain an increase in mAChRs associated with lowered 17β-estradiol levels during menopause or after ovariectomy. Since membrane estrogen receptors ERα, Erβ, and GPR30/GPER1 are co-expressed with mAChRs in cultured hippocampal neurons [75,76], it was hypothesized that the lack of estrogen receptor stimulation could also diminish the ACh release from the cholinergic neurons. As a result, low ACh levels could induce a compensatory increase in mAChRs expression [63,70]. On the contrary, estrogen replacement therapy applied immediately after ovariectomy may prevent the instantaneous upregulation of mAChRs observed in the rodent brain. It might be speculated that increased 17β-estradiol levels provided by the treatment may indirectly stimulate mAChRs gene expression by signalling through tyrosine kinase receptor (trkA) [70,77,78]. Indeed, 17β-estradiol can also influence the mAChRs expression levels by modulation of histone methylation via the mER-activated PI3K/Akt signal transduction pathway [79,80]. Prolonged periods of stress-increasing corticosterone reduced ACh uptake and increased the density of mAChRs in the hippocampus [81]. However, it is more likely that the changes in mAChRs expression levels are due to heterologous regulation by different hormones and transmitters [82].

## 5. Neurosteroids and Muscarinic Receptors in CNS

### 5.1. Cognitive Functions

A wide array of evidence suggests that cholinergic neurotransmission via mAChRs plays a key role in multiple cognitive functions [20] including attention [83], learning, and memory formation [84]. Antagonists of mAChRs are well known to cause confusion, disorientation, learning, and memory deficits which support an important role for the cholinergic system in these cognitive functions. Indeed, the administration of non-selective muscarinic receptor antagonist scopolamine has been and still is widely used as a valid pharmacological model of cognitive impairment [85] as well as an experimental compound increasing locomotor activity [86]. On the other hand, M_1_-preferring agonists, such as WAY-132983 [87] or xanomeline [20], or M_1_-selective partial agonist HTL0018138 [88] exhibit clear pro-cognitive effects. Attention, memory, and learning are essential for flexibility in responding to changing environments. These processes are highly interconnected as attention powerfully influences what we memorize [89]. Regarding the mechanisms of learning and memory, for relevant information to be acquired from the environment needs to be selected/filtered from many others, to be further encoded, stored, and/or retrieved (episodic memory), and it is regulated by attention. As for implicit memory, attention helps in sculpting what we learn, and working memory is closely related to selective attention at a particular moment [89,90]. Top-down attention (as well as working memory and inhibitory control) highly depend on activity within networks of prefrontal and parietal cortical areas [91,92].

The major role in cognition is attributed to M_1_ and M_3_ receptors. The M_1_ receptor is highly expressed in the cerebral cortex, hippocampus, and striatum, especially in extrasynaptic regions of glutamatergic pyramidal neurons [93]. Its localization is consistent with the cholinergic modulation of glutamatergic neurotransmission and thus neuronal excitability, synaptic plasticity as well as learning and memory processes [94,95,96]. Activation of M_1_ receptors was found to facilitate long-term potentiation (LTP) and enhance NMDAR activation [97]. The M_3_ receptor expression in CNS is much less abundant in comparison to M_1_. The M_3_-receptor knockout mice are deficient in hippocampus-dependent contextual fear conditioning [98]. Mice strain expressing the phosphorylation-deficient M_3_ receptor showed a deficit in fear conditioning, suggesting this type of learning depends on receptor phosphorylation/arrestin signalling pathway [99].

NSs improve memory by a mechanism involving modulation of key excitatory glutamatergic NMDA receptors and inhibitory GABAergic signals [100] as well as by affecting ACh release and regulating cholinergic neurotransmission [101,102,103]. The early studies on the influence of steroids on mAChRs function in cognition started with observations that 17β-estradiol upregulates the activity of the acetylcholine-synthesizing enzyme, choline acetyltransferase (ChAT) when administered after ovariectomy [104]. Further, it was reported that treatment with 17β-estradiol prevented ovariectomy-induced deficits in attention and learning [105] and 17β-estradiol was also shown to partially or completely reverse scopolamine-induced memory impairment in spatial memory tasks [62,106,107]. Since 17β-estradiol is needed for adequate functioning of basal forebrain cholinergic systems, a decrease in its levels may be responsible for menopause-associated cognitive decline.

The estrogen facilitates working memory via M_2_ receptors expressed in GABAergic neurons of the CA1 region of the hippocampus. Estrogen stimulates ACh release and thus activates these M_2_ receptors reducing GABAergic inhibitory tone on hippocampal pyramidal cells and increasing expression of NMDA receptors in the CA1 region of the hippocampus [108,109] resulting in the improvement of working memory [101]. The effects of 17β-estradiol on learning and memory are rather complex and involve not only muscarinic but also nicotinic and other receptors and signalling pathways [110]. It is disputable whether the improvement of cognition exerted by 17β-estradiol is due to the direct modulation of the M_1_ receptor alone. Typically, 17β-estradiol is present in the body at rather low concentrations (0.07–2.3 nM during the menstrual cycle and 22 nM during pregnancy). However, 17β-estradiol increases ACh efficacy at the M_1_ receptor only at concentrations of 100 nM and higher [13]. The levels of PROG are decreasing with age and participate in cognitive decline. It was demonstrated that PROG exerts opposing effects on memory during learning. High levels of PROG disrupt the acquisition phase of learning but facilitate the consolidation phase of memory formation [111]. Given that PROG increases the efficacy of ACh at the M_1_ and decreases it at M_2_ receptors, it might account for the improvement in the consolidation of memory formation. On the other hand, negative modulation of M_2_ receptors in GABAergic neurons may lead to an increased GABAergic tone and inhibition of NMDA receptors disrupting the acquisition phase of learning.

Adrenal stress hormones, such as cortisol (corticosterone in the rat), exert a major influence on brain functions, mainly related to stress reaction, memory, cognition, and sleep [112]. Specifically, glucocorticoids enhance the consolidation of new memories but impair information retrieval from long-term memory [113]. Corticosterone enhances the consolidation of memory associated with emotionally charged and stressful events [114,115]. This augmentation of glucocorticoid-induced memory consolidation is dependent on increased cholinergic activity within the basolateral amygdala (BLA) [116]. Moreover, in animal studies, increased levels of corticosterone within the hippocampus were associated with improved memory consolidation during the awake state, but not during sleep [117].

Stressful stimuli are associated with an increase in choline uptake and ACh release. The levels of glucocorticoids increase with prolonged stress or ageing [118], resulting in hippocampal neuronal loss and memory impairment. During chronic stress, elevated levels of Ach cause an up-regulation of mAChRs [81]. Conversely, a decrease in glucocorticoid levels stimulates hippocampal cholinergic neurotransmission [119], enhancing the process of memory consolidation [115]. In turn, an activation of the M_2_ (but not M_4_) receptor enhances corticosterone release [120]. Moreover, corticosterone increases Ach potency at the M_2_ but decreased it at the M_4_ receptor in functional experiments in vitro [13].

The PREG and sulphated neurosteroids includincludeding DHEAS, pregnanolone sulphate (PREGS), and estrone sulphate, also affect cholinergic neurotransmission. The PREG increases ACh release in many brain areas (frontal cortex, hippocampus, and amygdala) and improves spatial recognition and memory [102,103]. Sulphated neurosteroids reduced GABAergic recurrent inhibition in the hippocampus via enhancement of cholinergic neurotransmission [109]. The DHEAS and PREG reversed scopolamine-induced amnesia in rats and scopolamine-induced deficits in spatial memory in mice [121,122,123]. Moreover, inhibition of steroid sulfatase may be effective in improving memory by increasing levels of sulphated steroids in the brain [124]. Although sulphated NS improve cognitive processes related to muscarinic signalling, to the best of our knowledge, there is no evidence for the binding of sulphated NSs to mAChRs yet.

### 5.2. Alzheimer’s Disease

AD is a multifactorial progressive neurodegenerative disorder histologically defined by the presence of β-amyloid plaques in the brain. AD is associated with the gradual deterioration of memory and cognitive functions, impairment of cholinergic neurotransmission (especially in the cortex and hippocampus), and morphological changes in the brain (decrease in thickness of grey matter and increase in ventricular volume). Indeed, known factors associated with AD development (ageing, genetic factors, vascular diseases), there are many hypotheses concerning AD pathophysiology, including the formation of neurofibrillary tangles of hyperphosphorylated τ-protein inside the neurons (τ-hypothesis), abnormal extracellular deposits of already mentioned amyloid β (amyloid hypothesis), oxidative stress, inflammation, alterations in brain vasculature, cholinergic neuron damage, etc.

The amyloid hypothesis is tightly connected to the deterioration of cholinergic transmission. The amyloid hypothesis is based on the excessive accumulation of the extracellular deposits of beta-amyloid protein (Aβ) within the brain [125]. The Aβ peptide is the product of proteolytic cleavage of the transmembrane amyloid precursor protein (APP) by β- and γ-secretase. The APP is cleaved by these enzymes into several peptides which are 30–51 amino acids long. The most common forms are Aβ_40_ and Aβ_42_. The Aβ_42_ is most prone to oligomerisation. The Aβ_42_ oligomers are noxious inducing neuroinflammation in microglia cells and resulting in oxidative stress, and neurotoxicity, leading consequently to neurodegeneration [126]. Cleavage of APP by α-secretase precludes cleavage of APP by β-secretase resulting in the production of a soluble harmless non-amyloidogenic peptide (sAPP-α) which is responsible for the modulation of the normal neuronal excitability. Importantly, both pathways are regulated by the presence of membrane CLR [127]. Activation of the muscarinic M_1_ receptor leads to an increase in α-secretase activity both in vitro and in vivo [128,129,130]. Therefore, therapy by M_1_ agonists or PAMs was proposed [131]. In addition, activation of M_1_ was shown to reduce τ-protein phosphorylation, both in vitro and in vivo [132,133,134]. Mutations in the APP gene leading to increased cleavage by β-secretase are the cause of the familial form of AD that is characterized by early onset and fast progression. However, the familial form of AD accounts for less than 3% of AD cases. The majority of AD cases are sporadic forms of AD conditioned by many genetic, environmental, nutritional, and behavioural risk factors. The sporadic form of AD is late onset with slow progression. The major genetic risk factor of sporadic AD is the ε4 allele of apolipoprotein E (ApoE). The ApoE is a glycoprotein mediating lipid transport and endocytosis of CLR and lipoproteins. The ApoEε4 allele is associated with a high risk for AD as it promotes Aβ deposition and τ-protein hyperphosphorylation.

The cholinergic hypothesis of AD was built based on progressive neurofibrillary degeneration of limbic and neocortical cholinergic innervation associated with AD (Figure 2). Cholinesterase inhibitors (e.g., tacrine, donepezil, galantamine, rivastigmine) used as a treatment in AD patients, increase the availability of ACh at synapses and have been proven clinically relevant [135]. A defective coupling of neocortical M_1_ receptor to G-proteins is associated with cognitive decline in AD and dementia [22]. Additionally, it was suggested that M_5_ receptors of cerebral microvasculature are involved in the dilation of cerebral blood vessels [42]. Thus, the diminished cholinergic tone in AD, preventing proper vasodilatation, may play a part in the progression of AD. Therefore, stimulation of the M_5_ receptor may be beneficial in AD therapy [42].

As already mentioned, some neurosteroids (such as PROG) exert neuroprotective effects and depress the level of the hyperphosphorylation of τ-protein [52]. However, direct allosteric modulation of M_1_ signalling by 17β-estradiol at physiological concentrations is unlikely [13]. Therefore, the favourable effects of long-term estrogen therapy on cholinergic neurotransmission [136] (Table 3) are probably due to its influence on nicotinic acetylcholine receptors via estrogen receptor-β [137]. It might be also a result of an increased level of ACh due to activation of the GPR30 expressed mainly in the cholinergic neurons of the basal forebrain [76].

**Figure 2 ijms-24-00507-f002:**
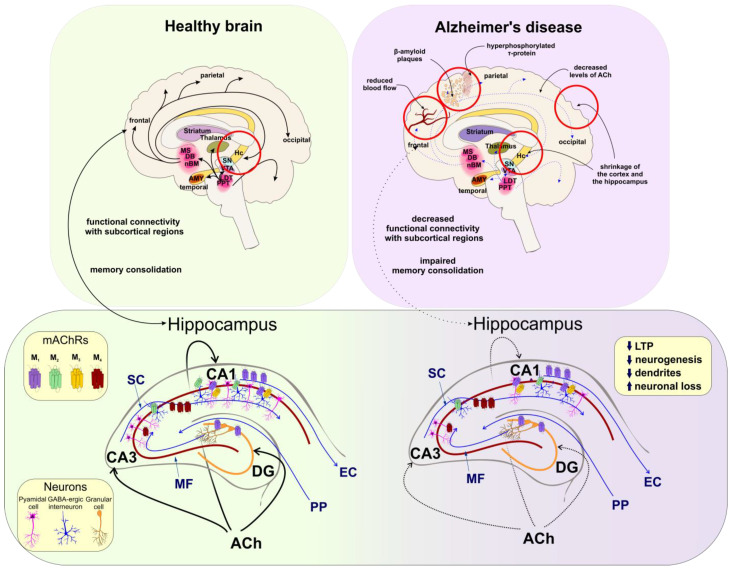
Changes in the human brain associated with AD. **Left:** In the healthy brain, projections from the basal forebrain and the brainstem (thick black arrows) supply cortical and subcortical brain regions (respectively) with ACh. Cholinergic neurons support cognitive functions and memory formation by sustaining functional connectivity between frontal cortical regions and the hippocampus (Hc) (black arrow). The Hc excitatory trisynapric circuit projections (blue lines) include pedunculopontine nucleus (PPN) projecting to the diagonal band of Broca (DG) (orange line-granule cells), DG projects to CA3 via MF, and CA3 region sends ith projections to CA1 via SC. (CA3 and CA1 are shown as a red line and pink-pyramidal cells). The CA1 further projects to EC. Activation of M_1_ (purple) mAChRs expressed mainly in granule cells of DG and M_1_/M_3_ (yellow) receptors at pyramidal neurons (pink) of the CA1, increases the activity of NMDARs (not shown) in long-term potentiation (LTP), synaptic plasticity, neurogenesis, memory encoding, and consolidation. The M_2_ (green) receptors are expressed mainly on interneurons (blue) and their activation reduces CA3 and CA1 activity. M_4_ (dark red) is expressed presynaptically on glutamatergic terminals. Their activation inhibits Schaffer collateral. **Right:** In cognitive decline and AD the ACh levels are decreased (thin blue arrows). Due to the accumulation of β-amyloid plaques, τ–protein tangles, neuronal loss in CA1, and reduced neurogenesis, the neocortex and hippocampus lose their volume. **Lower part:** Hc pathways are less active (thin blue arrows) resulting in impairment of memory. Stimulation of M_1_ in the DG, CA3, and CA1 as well as M_2_ on the GABA-ergic interneurons increases the excitability of pyramidal neurons in Hc and supports memory formation. According to [126,138,139].

**Table 3 ijms-24-00507-t003:** Effects of the steroids on the muscarinic signalling in cognitive decline and AD.

	M_1_	M_2_
17-β estradiol (mechanism independent from mAChRs) ↑ ACh levels ↑ LTP ↑ Learning acquisition phase ↑ Memory consolidation		↑ ACh potency ↓ Activation of GABA-ergic neurons (Hc) ↑ Excitability of pyramidal neurons (Hc) ↑ Activation of NMDARs ↑ LTP ↑ Learning acquisition phase ↑ Memory consolidation
Progesterone	↑ ACh efficacy ↑ LTP ↑ Learning acquisition phase ↑ Memory consolidation	↓ ACh efficacy ↑ Activation of GABA-ergic neurons (Hc) ↓ Excitability of pyramidal neurons (Hc) ↓ Activation of NMDARs ↓ Learning acquisition phase
Corticosterone		↑ ACh potency ↓ Activation of GABA-ergic neurons (Hc) ↑ Excitability of pyramidal neurons (Hc) ↑ Learning acquisition phase

Abbreviations: increase ↑, decrease ↓, LTP-long-term potentiation, hippocampus (Hc).

### 5.3. Schizophrenia

Schizophrenia is a chronic psychiatric disorder with prevalence mostly in adolescence and early adulthood. Schizophrenia with its complex neurobiological background is accompanied by three categories of symptoms: (i) “Positive symptoms” are behaviours that are not normally present (hallucinations, delusions, and confusion associated with recurrent psychosis). (ii) “Negative symptoms” are fading or no longer present behaviours (anhedonia, social isolation, poverty of speech, loss of motivation). (iii) “Cognitive symptoms” are manifested as cognitive dysfunctions (Figure 3).

The hallmark of schizophrenia is overactive dopaminergic pathways in the central nervous system, especially the mesolimbic area, resulting in positive symptoms [140]. On the other hand, the negative symptoms result from reduced D_1_ receptor activation in the prefrontal cortex [141] and decreased activity of the nucleus caudatus [142]. Negative symptoms that arise are thought to occur also as a result of overactivation of the prefrontal cortex. The imbalance between increased long-term potentiation (LTP) and deficits in long-term depression (LTD) is often caused by transient inhibition of NMDARs and accompanied by negative and cognitive symptoms of the disorder [25].

In general, the dopaminergic and cholinergic systems are deeply interconnected. The release of ACh in the striatum is stimulated through the release of dopamine [38]. The anticholinergic drugs cause retrograde amnesia, impaired general cognitive performance, and worsen psychosis but also cause a modest improvement of negative symptoms. The pro-cholinergic drugs, such as cholinesterase inhibitors, improve cognitive functions [38]. A commonly used antipsychotic clozapine has a unique property to antagonize D_2_ receptors and serotonergic 5-HT_2A_ and 5-HT_2C_ receptors and to activate M_4_ receptors. The active metabolite of clozapine, N-Desmethylclozapine (NDMC), is an M_1_-preferring agonist [143]. Striatal M_4_ receptors modulate behaviour. Thus, changes in cholinergic signalling in the striatum contribute to positive and negative symptoms of schizophrenia. The M_1_ and M_4_ receptors in the cortex and basal ganglia, M_2_ in the thalamus and brainstem, and M_5_ in the brainstem and midbrain (substantia nigra and the VTA) affect dopamine release [144].

The expression level of M_1_ and M_4_ receptors is lower in the cortex, hippocampus, and striatum but not in the thalamus of the post-mortem samples from schizophrenia patients [145,146]. Therefore, centrally acting muscarinic agonists or PAMs selective for M_1_ and/or M_4_ receptors could be efficacious in treating the positive (psychosis) and negative (cognitive decline) symptoms associated with schizophrenia [21,37,38]. In preclinical studies, the M_1_/M_4_-preferring agonist xanomeline [147,148] improved verbal learning and short-term memory function [149,150]. However, xanomeline was withdrawn from clinical studies due to the severe side effects [150]. The M_1_-preferring allosteric agonist TAK-071 improved cognitive deficits in a rodent model of schizophrenia, exerting fewer side effects than xanomeline [151].

Indeed, 17β-estradiol may play a protective role against the development of schizophrenia [152]. Negative symptoms are less manifested at high 17β-estradiol levels during the menstrual cycle and pregnancy [78]. Treatment with PREG and subsequent increases in levels of ALLO and PREGS reduced the severity of negative and positive symptoms [100,153]. Thus, steroidal PAMs of M_1_ and M_4_ receptors have therapeutic potential in schizophrenia. PREG has a high affinity for mAChRs [13], yet its detailed effect on mAChRs concerning schizophrenia needs to be further investigated.

**Figure 3 ijms-24-00507-f003:**
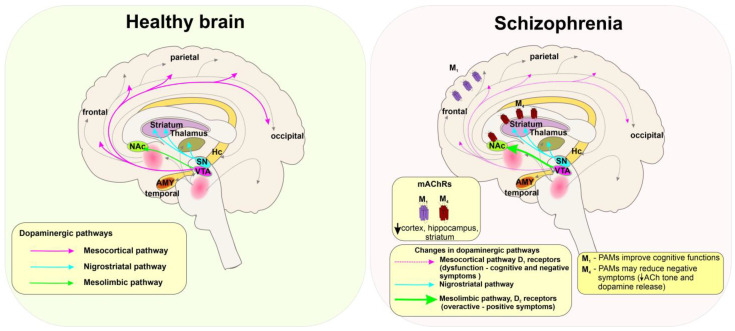
Changes in the human brain associated with schizophrenia. **Left**: In the healthy brain dopaminergic pathways arising from the ventral tegmental area (VTA) project into the neocortex (mesocortical pathway, pink), and to the nucleus accumbens (NAc) (mesolimbic pathway, green), while substantia nigra (SN) projects into the striatum and thalamus (nigrostriatal pathway, cyan). The distribution of dopaminergic projections greatly overlaps with cholinergic pathways (grey dashed lines). **Right:** In schizophrenia, dysfunction of the mesocortical pathway and decreased D_1_ receptors activation (dashed pink lines) are associated with negative symptoms. Overactive mesolimbic pathways (thick green lines) and increased activity of D_2_ receptors are related to positive symptoms of schizophrenia. Selective modulation of the central M_1_ and M_4_ receptors may represent the possible treatment of both negative and positive symptoms of the disease according to [144].

### 5.4. Seizures and Epilepsy

Overall, epilepsy is a collective name for a group of brain disorders consisting of a complex spectrum of different seizure types and syndromes. According to the International League Against Epilepsy (ILAE), a seizure is defined as “a transient occurrence of signs and/or symptoms due to abnormal excessive or synchronous neuronal activity in the brain.” Epilepsy is defined as a “disease characterized by an enduring predisposition to generate epileptic seizures and by the neurobiological, cognitive, psychological, and social consequences of this condition.” Therefore, a seizure is an event while epilepsy is a disease involving recurrent unprovoked seizures. Despite many advances in epilepsy research, presently an estimated 30% of people with epilepsy show drug resistance to currently available antiepileptic drugs (AEDs) [28]. A rather general explanation for the cause of seizure manifestation is a prolonged imbalance between the levels of excitatory and inhibitory neurotransmission (Figure 4).

Cholinergic neurotransmission plays an essential balancing role in brain excitability. Exposure to organophosphates (such as pesticides or chemical warfare agents) induces cholinergic neurotoxicity which may lead to the development of acute status epilepticus (SE), which may lead to the development of epilepsy [28]. As defined by the ILAE, “SE is a condition resulting either from the failure of the mechanisms responsible for seizure termination or from the initiation of mechanisms which lead to abnormally prolonged seizures. It is a condition that can have long-term consequences, including neuronal death, neuronal injury, and alteration of neuronal networks, depending on the type and duration of seizures”.

Enhancement of cholinergic activity by stimulation of mAChRs, mainly M_1_ receptor by pilocarpine, produces convulsions in laboratory rodents and is widely used as a model of SE in research laboratories [154]. Similarly, overstimulation of mAChRs following organophosphate poisoning may result in seizures [155]. An increase in ACh levels leads to an increase in susceptibility to seizures in experimental animals. Knockout mice models lacking acetylcholinesterase (AChE) are highly susceptible to seizures (progressing to tonic seizures) [156]. On the other hand, the mice with only partially eliminated AChE in the CNS by knocking out protein anchoring AChE to the membrane were not prone to develop spontaneous seizures and expressed a normal phenotype [157]. In accordance, brain-penetrating M_1_-selective antagonist VU0255035 suppresses the severity of the seizures and prevents SE after exposure to paraoxon or soman [158]. Repetitive seizure activity was shown to affect mAChRs expression throughout the brain. The expression levels of auto-inhibitory M_2_ receptors are higher in the neocortex of patients with focal temporal lobe epilepsy (TLE) [159] as well as in the brainstem of pentylenetetrazole-kindled epileptic rats [160] as the result of compensatory mechanisms. The expression levels of stimulatory M_3_ receptors are lower [161].

The importance of NS and steroid hormones in epileptogenesis was already described in detail [15,28,162]. From the wide range of steroidal compounds affecting susceptibility to seizures 17β-estradiol, PREGS, DHEAS, cortisol, and 11-deoxycortisol, were listed to have pro-convulsant effects. These NSs are acting mainly by negative modulation of GABA_A_ and/or positive modulation of NMDARs. Interestingly, NSs can regulate ACh levels and thus play a crucial role in epileptogenesis. For example, intraperitoneally administered DHEAS enhanced the release of ACh in the rat cortex and hippocampus [102,163]. PROG, pregnanolone, androsterone, etiocholanone, DOC, THDOC, and ALLO are potent allosteric agonists and modulators of the synaptic and extrasynaptic GABA_A_ receptors, enhancing phasic, and tonic neuronal inhibition [28].

A rise in the level of neuroprotective and antiseizure ALLO in the hippocampus has antiseizure effects [164]. Cyclic changes of estrogens and PROG are important in the pathogenesis of catamenial (menstrual cycle-related) epilepsy, as the changes in their levels correlate with the seizure incidence during the perimenstrual or periovulatory period [165].

The short-term selective stimulation of M_1_ or M_4_ receptors located mainly at glutamatergic terminals was shown to normalize hippocampal activity [166]. M_1_ receptors are responsible for tonic cholinergic excitation of CA1 and CA3 pyramidal neurons and M_4_ receptors modulate presynaptic inhibition of Schaffer collateral EPSPs [167]. The activation of M_1_ receptors enhances the excitability of fast-spiking putative (parvalbumin-positive) PV1 interneurons contributing to inhibitory tone within hippocampal circuitry [168]. Therefore, temporally-limited positive modulation of hippocampal M_1_ receptors may contribute to the anticonvulsive effect of PROG. Repetitive stress increases levels of corticosteroids resulting in an increased risk of seizures and epileptogenesis [169]. Consequently, long-lasting exposure to increased levels of corticosterone is often used as a model of chronic stress-inducing limbic epileptogenesis [28]. The M_2_ and M_4_ receptors are distributed both pre- and post-synaptically where expression of the M_2_ muscarinic subtype is characteristic of interneurons [93]. The pro-convulsant effects of corticosterone may be partially attributed to the positive modulation of the M_2_ and negative modulation of the M_4_ receptors [161].

**Figure 4 ijms-24-00507-f004:**
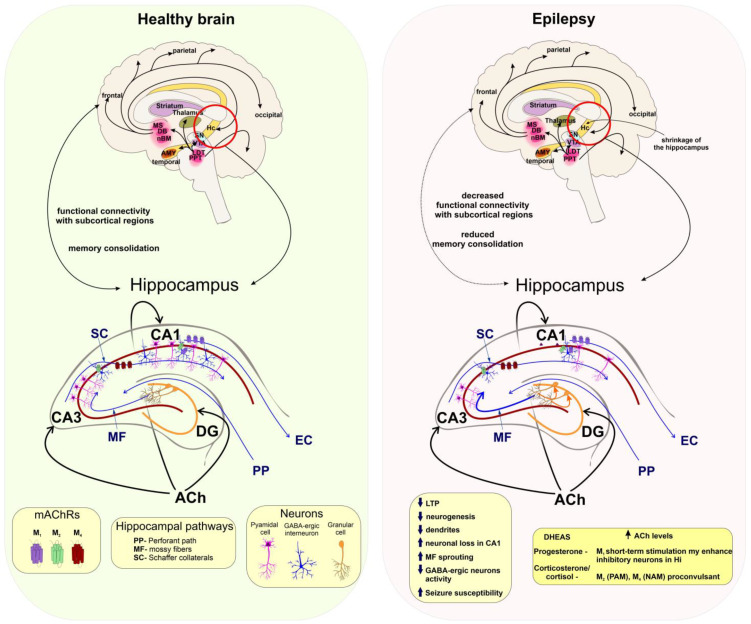
Cholinergic projections in the brain and hippocampal circuit in health and epilepsy. **Left**: In the healthy brain, cholinergic neurotransmission provides functional connectivity between the neocortex and hippocampus (Hc) and thus supports memory consolidation. In the Hc, Ach maintains a balance between excitation and inhibition within neuronal pathways involved in learning and memory (see Figure 2). Cholinergic projections regulate Hc activity within the dentate gyrus (DG), CA3, and CA1 regions of the Hc and M_1_ receptors (purple) regulate excitability mainly of the pyramidal neurons (pink) and are also expressed on GABA-ergic interneurons (blue) in the CA1. The M_2_ receptors (green) are expressed mostly presynaptically on GABA-ergic interneurons and their activation supports the CA1 excitability. The M_4_ receptors on are expressed on glutamatergic neurons of SC and their activation decreases the activation of CA1. **Right:** In the epileptic brain, Hc circuits are affected by MS sprouting and increased excitability within DG granule cells. Mossy fibres (MF) send more projections to the CA3 and further increases Hc excitability and seizure susceptibility. These eventually lead to neuronal loss in the CA1 and impairment of Hc functions. Short-term stimulation of M_1_ expressed on interneurons may reduce excitability within CA1. M_4_ PAMs may inhibit overactive SC and thus CA1 activity. According to [45].

### 5.5. Parkinson’s Disease

Parkinson’s disease (PD) is associated with dopamine deficiency caused by the loss of dopaminergic neurons within the basal ganglia (BG) structure, substantia nigra pars compacta (SNc), and widespread intracellular protein (α-synuclein) accumulation (Lewy bodies). Decreased dopamine levels in the substantia nigra are responsible for slowness of movements, body rigidity, and rest tremors [170] (Figure 5).

Cholinergic neurotransmission mediated by mAChRs (mainly M_4_ and M_1_) and nAChRs is known to regulate dopamine release from the terminals originating from the SNc with the M_4_ receptor playing a central role [34]. In the striatum, mAChRs and dopaminergic receptors are often colocalised on Substance P-positive neurons, where the M_1_ receptor is co-expressed with the D_2_ receptor in stimulating indirect pathways, while the M_4_ receptor is co-expressed on medial spiny neurons with the D_1_ receptor in spiny projection neurons (dMSN) that comprise the BG direct pathway. Activation of M_4_ receptors reduces the glutamatergic tone in the striatum. Moreover, activation of M_4_ on dMSN leads to a decrease in dopamine release from dopaminergic terminals in SNc. On the cholinergic interneurons activation of M4 receptors may result in decreased tonic firing [173,174]. Thus, selective negative modulation of M_4_ receptor signalling may be effective in disorders accompanied by reduced dopamine levels (Table 1 and Figure 5). Indeed, NAMs of the M_4_ receptor can improve symptoms of Parkinson’s disease by increasing dopamine release [25,34]. On the other hand, centrally acting M_1_ antagonists such as trihexyphenidyl, pirenzepine, or VU0255035 alleviate symptoms of striatal dystonia [175].

NSs exert neuroprotective effects related to PD. In the animal model of PD, PROG, and 17β-estradiol protected against the degeneration of dopaminergic neurons induced by neurotoxin MPTP (1-methyl-4-phenyl-1,2,3,6-tetrahydropyridine) [176]. Interestingly, the levels of neuroprotective NS, such as PROG and ALLO, are lowered in the cerebrospinal fluid of PD patients [177]. Some of the beneficial effects of NSs on PD might be attributed to mild negative modulation of the M_4_ receptors by 17β-estradiol and PROG as PROG slightly decreases the efficacy of ACh, and 17β-estradiol decreases its potency at this receptor [13].

### 5.6. Substance Abuse

According to the Diagnostic and statistical manual of mental disorders (American Psychiatric Association, DSM-5 2013), substance abuse has several stages, starting with acute drug use and intoxication. Subsequent compulsive and harmful use may develop into dependence and withdrawal syndrome (with or without delirium) and drug-seeking behaviour that may trigger amnesic syndrome, residual and late-onset psychotic disorders, and other mental and behavioural changes. Addiction may be defined as a chronic illness characterized by compulsive drug seeking and use. Overall, substance abuse and addiction are associated with the function of dopamine. Dopamine cell bodies located in the ventral tegmental area (VTA) with their projections to the nucleus accumbens (NAc) and prefrontal cortex (PFC) represent the midbrain reward circuitry [178]. Stimulation of dopamine release within NAc, a region in the basal forebrain (part of the ventral striatum) which is activated by food and water intake, but also by psychomotor stimulants and opiates, produces the feeling of pleasure, euphoria, and reward. By modulation of the various brain regions interconnected with NAc, such as VTA, PFC, hippocampus, and amygdala, dopamine affects brain activity and establishes a mechanism of learned behaviour associated with reward [179] (Figure 6). Addiction develops because of alterations in these reward circuits.

Profound sex differences in course, symptoms, and treatment of substance use disorders indicate that estrogens and PROG are involved in mediating responses to drugs of abuse [180]. Typically, women are more vulnerable than men to the deleterious consequences of drug use at every phase of the addiction process. Exogenous PROG protects against the reinforcing effects and attenuates drug cravings [181]. On the other hand, estrogen enhances the reinforcing effects [182].

**Figure 6 ijms-24-00507-f006:**
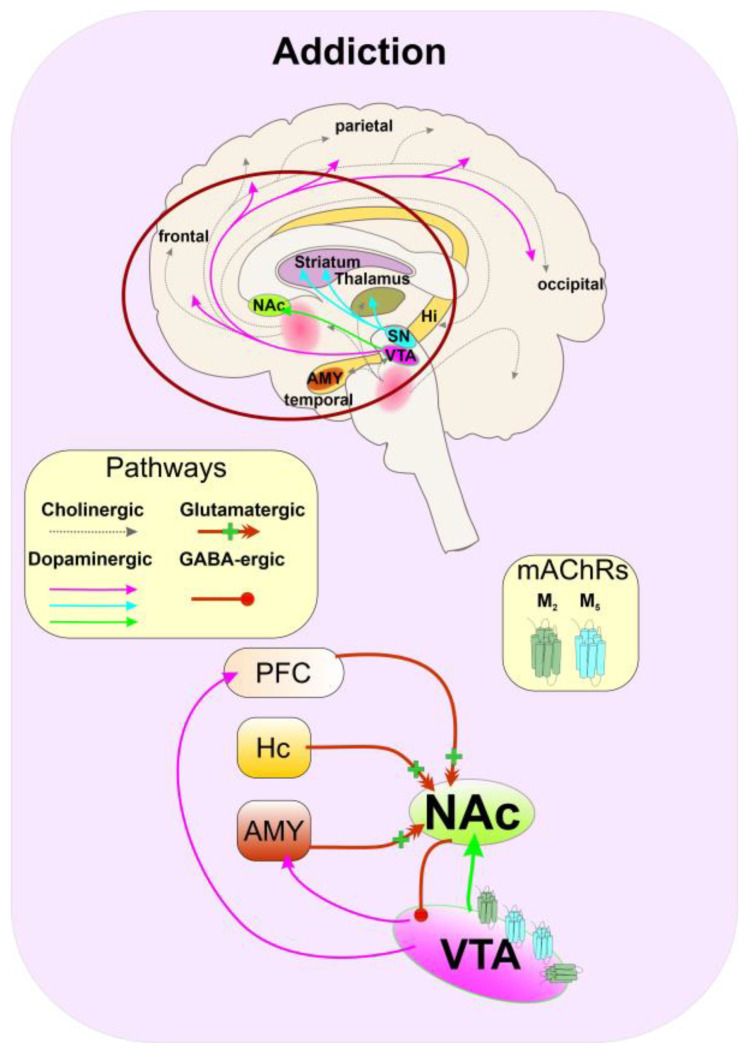
Schematic representation of the pathological dopaminergic circuitry in addiction. Repetitive rewarding stimulus induces excessive dopamine release from the ventral tegmental area (VTA) projections (pink lines) to the prefrontal cortex (PFC), amygdala (AMY), and most importantly to medial spiny neurons (MSNs) of nucleus accumbens (NAc) (light green), involved in the direct and indirect motor pathways (see Figure 5). Learned behaviour strengthens excitatory projections (red lines) from AMY, hippocampus (Hc), and PFC to NAc and supports addictive behaviour. MSNs of NAc containing D_1_ receptors project back to the VTA and inhibit dopamine release. Inhibition or negative modulation of M_5_ in the VTA might decrease dopamine release and prevent the further development of addiction according to [179,183].

At least since the “opioid wars” in China, muscarinic antagonists atropine and scopolamine are used to prevent opiate overdose and the development of addiction, as well as support detoxification, indicating the involvement of mAChRs in the process of addiction and substance abuse. Dopamine release within NAc is decreased in M_5_ receptor knock-out mice. These are less sensitive to the rewarding effects of cocaine, indicating the role of M_5_ receptors in the regulation of the dopaminergic reward circuitry [184]. In addition, the NAM of the M_5_ receptor ML375 [185] affected cocaine-related behaviour without impairing behaviour not associated with drug use [25,184]. Probably, a decrease in potency and efficacy of ACh at the M_5_ receptor by pregnanolone [13] contributes to its beneficial effects on rewarding effects in drug abuse. The M_2_ receptors are expressed in the pre- and post-synaptic sites in the VTA. Their activation may affect cognitive and goal-based dopamine-mediated behaviour. Negative allosteric modulation of M_2_ receptors by pregnanolone may further diminish rewards seeking behaviour.

### 5.7. Depression

By the definition, depression is a mental state of low mood and apathy. Depression affects a person’s feelings, behaviour, and motivation. Major depressive disorder (MDD) is a multifactorial syndrome that involves disruption of mood, cognition, and other processes, including sleep, appetite, and libido. Signs and symptoms of MDD are mainly linked to low levels of brain monoamines dopamine, serotonin, and noradrenaline. In addition, stress and augmented glutamatergic neurotransmission often lead to neurodegenerative changes [186]. Changes in hippocampal neurogenesis, often associated with chronic stress in animal models may be in part mediated by cholinergic dysfunction, which in turn could underlie the cognitive disturbances observed in depression [187].

Antidepressant treatment involves the use of selective reuptake inhibitors of serotonin (SSRI), noradrenaline (NRI), both serotonin and noradrenaline (SNRI) as well as tricyclics and monoamine oxidase inhibitors (MAOI) which regulate and increase monoamine. Pharmacological targeting of the cholinergic system may represent a viable alternative. Muscarinic agonists (e.g., arecoline, oxotremorine), and AChE inhibitors (e.g., physostigmine, donepezil) provoke symptoms of depression. Stress-increased ACh release within the forebrain and activation of the septohippocampal pathway also results in MDD symptoms. Additionally, cholinergic dysfunctions may account for the development of cognitive symptoms associated with MDD [187]. The expression of M_2_ and/or M_4_ receptors are decreased in the dorsolateral prefrontal cortex of MDD patients [188]. The single-nucleotide polymorphism of A->T 1890 in the 3′ UTR of the (CHRM2) gene coding for M_2_ is gender-specific and characteristic of women with MDD [30].

In human studies, antidepressant-like properties of muscarinic antagonist scopolamine were first recorded about 30 years ago during the research of scopolamine as an anti-abuse drug on sleep and mood of alcoholics [189]. Unlike classical antidepressants, scopolamine exerts rapid antidepressant effects within a few days. Although scopolamine is a non-selective muscarinic antagonist, the anti-depressant mechanism of action of scopolamine was found to be dependent on the presence of M_1_ and M_2_. The knockout (KO) mice lacking these receptors expressed diminished scopolamine effectiveness in reducing immobility time in the forced swim test. In addition, the effects of M_1_-preferring antagonist were mitigated in M_1_ KO mice but not in the mice lacking M_2_ [190]. In the same study, the M_2_-preferring antagonist SCH226206 was found to be ineffective in M_2_ KO mice. Inhibition of M_1_ receptors in the medial prefrontal cortex (mPFC) was found to play a major role in the rapid antidepressant effects of scopolamine [191].

As the anxiety and symptoms of low mood are temporally linked to events of reproductive transitions in women (menstrual cycle, pregnancy and postpartum, and perimenopause) it is presumed that changes in the mood are rather associated with fluctuations in the levels of estrogens and PROG than by their absolute concentrations [192]. Additionally, women are more likely than men to show an antidepressant response to scopolamine [193]. These findings indicate the interplay between sex hormones and mAChRs in MDD.

PROG is well known for its negative impact on mood and memory. Changes in plasma levels are tightly associated with symptoms of depression or premenstrual dysphoric disorder [194]. These effects of PROG were found to occur independently from the activation of the endocrine PROG receptor. Acute administration of low doses of PROG (which reliably elevates ALLO and PREG levels) had negative effects on mood, while higher doses had no effect [194]. PROG may contribute to the symptoms of depression via modulation of mAChRs by increasing ACh efficacy at M_1_ receptors while diminishing it at M_2_ [13]. Positive modulation of post-synaptic M_1_ and negative modulation of pre-synaptic auto-inhibitory M_2_ receptors may have a synergistic effect.

As already mentioned, 17β-estradiol increases cholinergic neurotransmission and affects the function of mAChRs enhancing cognitive functions and alleviating scopolamine-induced impairments in attention and memory tasks [136]. It was found that treatment with 17β-estradiol alleviates depressive symptoms in premenopausal women. Under conditions where levels of estrogens are fluctuating and decreasing more rapidly than under normal circumstances, hormone replacement therapy stabilizes 17β-estradiol levels and thus moderates the negative effects of fluctuations of 17β-estradiol levels [195] and 17β-estradiol reduces ACh efficacy at M_2_ receptors [13] which may lead to augmented cholinergic transmission due to a decrease in M_2_-mediated auto-inhibition of ACh release. These findings indicate that increase in Ach levels by 17β-estradiol contributes to MDD.

## 6. Conclusions

The importance of cholinergic neurotransmission and its widespread effects exerted by mAChRs in health and disease is unquestionable. Due to their expression in the CNS and their contribution to the pathology of numerous psychiatric and neurologic diseases and disorders, mAChRs represent possible therapeutic targets. Keeping in mind that the orthosteric binding sites of all five subtypes of mAChRs are structurally the same, selective activation or inhibition of a single subtype by an orthosteric ligand is virtually impossible. Pharmacological targeting of mAChRs by orthosteric ligands is accompanied by more or less severe side effects. The answer to the need for subtype-specific compounds has emerged and is represented by allosteric modulators that bind to the less conserved sites on mAChRs. Recent advances have greatly expanded the set of known allosteric modulators of mAChRs. Cholesterol allosterically modulates many GPCRs including mAChRs. The CLR-binding site differs from the common allosteric binding site on mAChRs located between the second and third extracellular loops and is oriented towards the membrane. Compounds sharing the steroidal scaffold with CLR, such as SHs and NSs, have the potential to allosterically modulate GPCRs. Most recently, it was revealed that corticosterone and PROG, allosterically modulate the functional response of mAChRs at physiologically relevant concentrations. This important finding, not only pointed at the steroids exerting their action via a non-genomic mechanism but also suggested that already well-known beneficial effects of treatments with steroids (17β-estradiol, PROG, corticosterone) may be contributed by allosteric modulation not only of NMDA and GABA receptors but also mAChRs. Modulation of mAChRs by SHs and NSs contributes to various CNS-related disorders and diseases. Namely: Allosteric modulation of M_1_ and M_4_ receptors may have a protective effect in the development of schizophrenia. Proconvulsant effects of corticosterone may be partially attributed to positive modulation of the M_2_ and negative modulation of the M_4_ receptors. Negative modulation of the M_4_ receptors PROG and 17β-estradiol may alleviate PD symptoms. Negative modulation of the M_5_ receptor by pregnanolone may diminish the rewarding effects of drug abuse. Negative modulation of the M_2_ by 17β-estradiol increasing level of ACh may contribute to MDD. Further, positive allosteric modulation of M_1_ receptors by NSs has therapeutic potential in cognitive deficits and the development of AD.

## 7. Perspectives

Pharmacotherapeutic targeting of mAChRs in the CNS is an ongoing challenge. In general, NSs pass the blood-brain barrier. Some of them exert a high affinity for CLR-binding sites at mAChRs. Thus, allosteric modulation of mAChRs via the CLR-binding site/sites represents a new approach for the development of potent and efficacious compounds with therapeutic potential in diseases and disorders associated with altered muscarinic signalling. Development of novel neuroactive steroids acting as allosteric modulators of mAChRs may be expected soon.

## Figures and Tables

**Figure 1 ijms-24-00507-f001:**
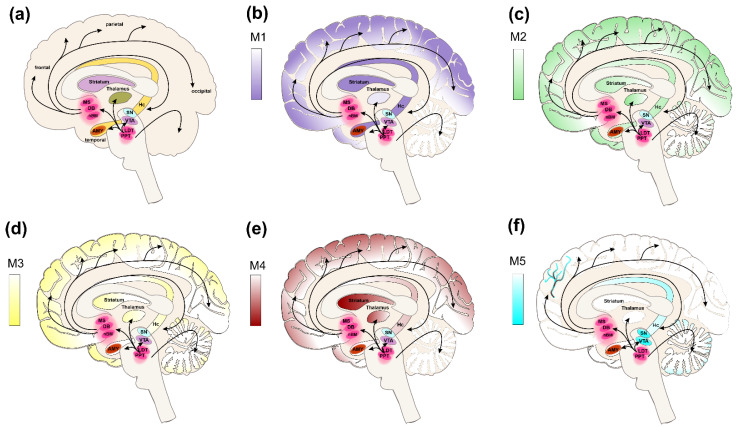
Cholinergic pathways and expression levels of muscarinic receptors in the human brain. (**a**) The cholinergic inputs to cortical and subcortical structures from the basal forebrain projections, including medial septum (MS), diagonal band of Broca (DB), and nucleus basalis magnocellularis (nBM) with projections (black arrows) to the neocortex (frontal, parietal, occipital), amygdala (AMY), and hippocampus (Hc). The brainstem cholinergic projections from the pedunculopontine nucleus (PPN) and laterodorsal tegmentum (LDT) to the thalamus and the structures of the basal forebrain. The distribution of individual muscarinic receptors subtypes (**b**) M_1_ (purple), (**c**) M_2_ (green), (**d**) M_3_ (yellow), (**e**) M_4_ (red), (**f**) M_5_ (cyan) with expression levels indicated by colour gradients. According to [45].

**Figure 5 ijms-24-00507-f005:**
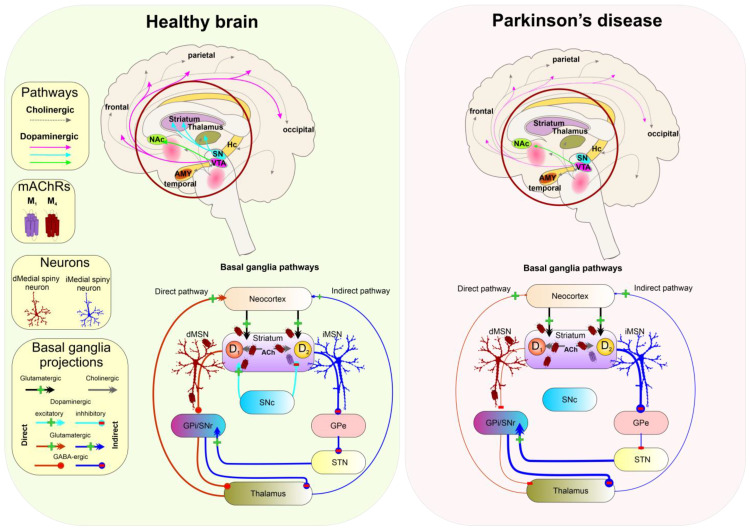
Neural pathways and circuits implicated in Parkinson’s disease. **Left:** In the healthy brain, the diagonal band of Broca (BG) nuclei provides control over the movements by the direct or indirect inhibition of the excitatory activity of the thalamus by the globus pallidus (GP). The net effect of activation of the direct pathway is a fine-tuned stimulation of the motor cortex, allowing for voluntary motor activity. Excitatory projections from the neocortical cholinergic interneurons (grey arrows) into the striatum and release of dopamine from the substantia nigra pars compacta (SNc) (cyan) activate striatal D_1_ (excitatory) receptors, which in turn activate inhibitory medial spiny neurons of direct pathway (dMSNs) (dark red). The dMSNs send their projections (red line) to the GPi/SNr and reduce their strong inhibitory tone. Though GPi/SNr send their inhibitory projections (red line) to the thalamus, the GP/SNr inhibitory tone is diminished, allowing for adjusted stimulation of the motor cortex (light red arrow) and controlled movement. The indirect pathway acts along with the direct pathway to prevent overstimulation of the thalamus and thus excessive or involuntary movements. In the indirect pathway, excitatory inputs from the cortex that synapse on cholinergic interneurons and dopamine release from the SNc, stimulate striatal dopaminergic D_2_ (inhibitory) receptors. This reduces the inhibitory tone of the iMSN projections (blue) on the GPe. Disinhibited GPe sends fewer inhibitory signals to STN. In turn, activated STN sends excitatory projections (blue arrow) to the GPi/SNr which stimulates it to send more inhibitory signals to the thalamus which sends fewer excitatory inputs to the cortex. Activation of the indirect pathway results in inhibition of the motor activity. **Right:** In the PD, loss of dopaminergic neurons projecting from the SNc results in diminished activation of the dMSN and disinhibition of iMSN, stimulating indirect pathway, resulting in strong inhibition of GPe. Consequently, strong activation of GPi/SNr and inhibition of the thalamus and cortex occur. In the effect, voluntary motor control is lost. Selective negative modulation of the M_4_ receptors on the corticostriatal projections, dMSN, and cholinergic interneurons (red receptor) could reduce parkinsonian motor disability according to [34,171,172].

**Table 1 ijms-24-00507-t001:** Signalling, expression sites, and implications in CNS disorders.

	Response	Expression	Activation	Inhibition	Implications in CNS Disorders
M_1_	↑ PLC, IP_3_, DAG, Ca^2+^ and PKC; depolarization and excitation (EPSPs)	neocortex, hippocampus, striatum [18,19]	↑ neuronal depolarization [31], ↑ learning and memory [20], ↓ dopamine release, locomotion, ↑ wakefulness, ↓ delta sleep, ↑ seizure activity [27]	↓ REM sleep [46] ↓ dopamine release	AD [22,23], schizophrenia [24,25], PD [39], cognitive dysfunction [24,25,38], seizures/epilepsy [27,28]
M_2_	↓ cAMP, ↑ GIRKs, ↓ VGCCs, neuronal hyperpolarization	hippocampus, cortex, olfactory bulb, basal forebrain, thalamus, cerebellum [19,41]	↓ neurotransmitter release, tremor, hypothermia, analgesia [26]	↑ locomotor activity, ↑ vasomotor centre, ↓ pituitary, ↓ food intake, ↓ growth hormone, ↓ prolactin, [26]	pain management [29], depression [30]
M_3_	↑ PLC, IP_3_, DAG, Ca^2+^ and PKC; depolarization and excitation (EPSPs)	cortex, basal ganglia, cerebellum [19,41]	↑ learning, memory [31]	disinhibition of dopamine release [47]	
M_4_	↓ adenylyl cyclase, ↓cAMP, ↑ GIRKs, ↓ VGCCs, hyperpolarization	striatum, neocortexhippocampus basal ganglia [19,24,32]	↓ neurotransmitter release, analgesia, ↓ dopamine release [34], anti-psychotic effects [36]		schizophrenia [21,24,25,37,38], PD [34,39], pain management [40]
M_5_	↑ PLC, IP_3_, DAG, Ca^2+^ and PKC; depolarization and excitation (EPSPs); ↑ PLD2	VTA, SN, brain microvasculature, cerebellum [19,24,31,41]	↑ cerebral vasodilation, ↑ dopamine release [26,27]; ↑ drug-seeking behaviour and reward [25]	↓ cerebral vasodilation [27] ↓ drug-seeking behaviour and reward [25]	AD [42,43], schizophrenia [25], substance abuse [43,44]

Abbreviations: AC, adenylyl cyclase; AD, Alzheimer’s disease; DAG, diacylglycerol; EPSP, excitatory postsynaptic potential; GIRK, inwardly rectifying K^+^ channels; IP_3_, inositol 1,4,5-trisphosphate; NO, nitric oxide; PD, Parkinson’s disease; PLC, phospholipase C; PLD2, phospholipase D_2_; PKC, protein kinase C; SN, substantia nigra; VGCC, voltage-gate Ca^2+^ channel; VTA, ventral tegmental area; ↑- activation and/or increase; ↓-inhibition and/or decrease.

**Table 2 ijms-24-00507-t002:** Effects of steroids on the expression level of mAChRs.

Treatment	Brain Area	Subtypes	Methods	Post-Ovariectomy Time	Duration of Hormone Treatment	Effect	Ref.
proestrus stage (high 17β-estradiol level)	rat cerebral cortex	all	[^3^H]NMS binding			↑	[64]
diestrus stage (low 17β-estradiol level)						↓	
EB and PROG replacement after ovariectomy				7 days		↑	
EB replacement after ovariectomy	rat medial basal hypothalamus	all	[^3^H]QNB binding		2 days	↑	[67]
EB replacement after ovariectomy	rat medial basal hypothalamus	all	[^3^H]QNB binding	14 days		↑	[68]
	rat medial preoptic area					↓	
17β-estradiol deprivation (ovariectomy)	rat hippocampus, frontal cortex, hypothalamus	M_4_	autoradiography	10 days		↑	[63]
17β-estradiol replacement				2 days	10 weeks	↓	
progesterone						↑	
17β-estradiol deprivation (ovariectomy)	rat hippocampus	all	[^3^H]QNB binding	2, 10, 15 days		↑	[69]
EB replacement				15 days	7 days	0	
				immediate treatment	21 days	↓	
17β-estradiol deprivation (ovariectomy)	rat hippocampus	M_1_–M_5_	immuno-precipitation	15 days		↑	[70]
EB replacement				15 days	7 days	0	
				immediate treatment	21 days	↓	
17β-estradiol deprivation (ovariectomy)			[^3^H]QNB binding	15 days		↑	
EB replacement				15 days	7 days	0	
				immediate treatment	21 days	↓	
17β-estradiol deprivation (ovariectomy)	rat amygdala, caudate putamen, dorsal hippocampus, motor cortex, retrosplenial cortex, ventromedial hypothalamus	M_1_, M_2_/M_4_, M_3_	autoradiography	10 days		0	[72]
long-term estrogen therapy	human left striatum, hippocampus, lateral frontal cortex, thalamus	M_1_ and M_4_	(SPET)			↑	[74]

The effect of 17β-estradiol and progesterone on the expression of mAChRs in the brain. Abbreviations: EB, 17β-estradiol benzoate; SPET, single photon emission tomography; ↑, increase; ↓, decrease; 0, no change.

## Data Availability

Not applicable.

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
