# Peer review of "Modulation of Muscarinic Signalling in the Central Nervous System by Steroid Hormones and Neurosteroids"

_ijms, 2022, doi:10.3390/ijms24010507_

Round 1

Reviewer 1 Report

The review by Szczurowska et al. is focused on the modulation of muscarinic receptor function by steroid hormones and neurosteroids. Although the abstract promises a review of allosteric modulation by neurosteroids and steroid hormones, this topic is mentioned only marginally.  If the authors would dedicate one chapter or subchapter to this topic I believe that it significantly improves the manuscript. In other aspects, the review is well and concisely written and the Figures are well-designed. 

Other points are rather minor:

1) line 46: board should be read broad, instead.

2) lines 86-87: the physiological role of M4 (locomotion, biological rhythm, pain modulation) is not mentioned, the authors directly focused on diseases (schizophrenia). Please add.

3) lines 87-88: the expression of M5 muscarinic receptors is low (it should be stressed) and other sources (Lebois et al. Neuropharmacology,136, 362, 2018) determined also other areas of M5 expression in the CNS.

4) line 115: it is not appropriate to start the chapter "Genomic effects" with the words "Non*genomic".

5) line 199: scopolamine is also experimentally used as a drug increasing locomotion.

6) Figure 2: in the legend, there are two M2 muscarinic receptors (green and yellow which should be M3). Red (M4) is on my display rather brown.

7) lines 443 and the following: this part should be completed with respect to AchE KO and AChE CNS KO (PRiMA KO) specific mice and their susceptibility to seizures.

Author Response

We thank the reviewer for constructive criticism. We addressed raised points as follows (blue).

The review by Szczurowska et al. is focused on the modulation of muscarinic receptor function by steroid hormones and neurosteroids. Although the abstract promises a review of allosteric modulation by neurosteroids and steroid hormones, this topic is mentioned only marginally.  If the authors would dedicate one chapter or subchapter to this topic I believe that it significantly improves the manuscript. In other aspects, the review is well and concisely written and the Figures are well-designed.

It is understood throughout our review that the modulatory effects of steroids are either genomic or allosteric. We have changed “allosteric modulation” to “modulation” to avoid confusion. In the Introduction, we give basic terminology and description of allosteric modulation and refer readers for further details to our recent review on direct allosteric modulation of muscarinic receptors by steroids (Ref 12).

Other points are rather minor:

1) line 46: board should be read broad, instead.

We corrected the typo.

2) lines 86-87: the physiological role of M4 (locomotion, biological rhythm, pain modulation) is not mentioned, the authors directly focused on diseases (schizophrenia). Please add.

We have expanded the paragraph on the role of M4 receptors in locomotion and rhythmicity.

3) lines 87-88: the expression of M5 muscarinic receptors is low (it should be stressed) and other sources (Lebois et al. Neuropharmacology,136, 362, 2018) determined also other areas of M5 expression in the CNS.

We expanded the paragraph on the expression of M5 receptors and amended Table 1 accordingly.

4) line 115: it is not appropriate to start the chapter "Genomic effects" with the words "Non*genomic".

We reworded the beginning of the section.

5) line 199: scopolamine is also experimentally used as a drug increasing locomotion.

We have amended the paragraph as suggested.

6) Figure 2: in the legend, there are two M2 muscarinic receptors (green and yellow which should be M3). Red (M4) is on my display rather brown.

We have updated Figure 2 and its caption.

7) lines 443 and the following: this part should be completed with respect to AchE KO and AChE CNS KO (PRiMA KO) specific mice and their susceptibility to seizures.

We have added a discussion of AChE KO mice relating to epilepsy and seizures.

Reviewer 2 Report

I have carefully read the manuscript (Manuscript ID, ijms-2094756) entitled “Modulation of muscarinic signalling in the central nervous system by steroid and neurosteroids”. The authors review effects of steroid hormones and neurosteroids on muscarinic signalling in the central nervous system (CNS). The manuscript is well written, and the Figures and Tables are presented clearly. This review article will be of interest to both neuroscientists and endoclinologists.

The following is my advice for further improvement of the manuscript (not criticism).

As the authors mentioned, nuerosteroids are steroids synthesized de novo from cholesterol in the CNS. The locally synthesized steroids can have different physiological roles from circulating steroid hormones. I think that the authors had better discuss distinct characteristics of neurosteroids on muscarinic signalling from those of circulating steroid hormones.

Author Response

We thank the reviewer for suggestions. We addressed them as follows (blue).

As the authors mentioned, nuerosteroids are steroids synthesized de novo from cholesterol in the CNS. The locally synthesized steroids can have different physiological roles from circulating steroid hormones. I think that the authors had better discuss distinct characteristics of neurosteroids on muscarinic signalling from those of circulating steroid hormones.

We have added paragraph (Ln 67-71) on the concentration of circulating steroids and locally synthesised neurosteroids.

Round 2

Reviewer 1 Report

The authors improved the manuscript. It could be accepted in the present form.